# Single-shot magnon interference in a magnon-superconducting-resonator hybrid circuit

Moojune Song [1,2], Tomas Polakovic [3], Jinho Lim [4], Thomas W. Cecil [5], John Pearson [1,6], Ralu Divan [6], Wai-Kwong Kwok[1], Ulrich Welp[1], Axel Hoffmann [4] ✉, Kab-Jin Kim [2] ✉, Valentine Novosad [1] ✉ & Yi Li [1] ✉

Magnon interference is a hallmark of coherent magnon interactions. In this work, we demonstrate single-shot magnon interference using up to four magnon pulses in two remotely coupled yttrium iron garnet spheres mediated by a coplanar superconducting resonator. By exciting one YIG sphere with injected microwave pulses, we achieve coherent energy exchange between the two spheres, facilitating their interference processes, including Rabi-like oscillation with a single pulse, constructive and destructive interference with two pulses, and interference peak sharpening with up to four pulses—analogous to diffraction grating in optical interference. The resulting interference patterns can be precisely controlled by changing the frequency detuning and time delay of the magnon pulses. The demonstration of time-domain coherent control of remote magnon interference opens new pathways for advancing coherent information processing through multi-operation, circuit-integrated hybrid magnonic networks.

Interference is a fundamental phenomenon in physics where two coherent waves combine constructively or destructively depending on their phase difference. This effect has been widely observed across various types of waves, such as light, acoustic waves, spin waves, or even gravitational waves. When waves are spatially confined within a resonator or a cavity, their interaction are significantly enhanced due to multiple reflections in the resonator, leading to cavity-enhanced interference phenomena with examples of mode hybridization, vacuum Rabi oscillation, and Ramsey interference. These effects have played a crucial role in establishing new protocols for coherent information processing and transduction, with direct applications in quantum computing[1,2].

The emerging field of cavity magnonics combines the advantages of cavity-enhanced interactions with the versatility of magnons for exploring novel functionality based on magnon interference.

Propagating magnons have been widely explored for interference-based information processing, with examples of spin wave logic gates[3–5], directional couplers[6,7], and multiplexing operations[8,9]. These functionalities rely on the interference of two or more propagating magnon waves[3–5,10–13], where constructive or destructive interference is used for performing logic operations. However, fundamental limitations such as the short magnon propagation distance and inefficient magnon excitations have hindered the advancement of coherent magnon spintronics. Cavity magnonics[14–17] provides a solution to these spatial and efficiency constrains in coherent magnonic information processing. By achieving strong coupling between magnons and microwave photons[18–27], high-fidelity coherent transduction between magnons and photons can be achieved, allowing for flexible state control within the magnon relaxation time. This cavity-confined interference process has been applied for designing phase coherent

[1]Materials Science Division, Argonne National Laboratory, Lemont, IL 60439, USA. [2]Department of Physics, Korea Advanced Institute of Science and Technology, Daejeon 34141, Republic of Korea. [3]Physics Division, Argonne National Laboratory, Lemont, IL 60439, USA. [4]Department of Materials Science and Engineering and Materials Research Laboratory, Grainger College of Engineering, University of Illinois Urbana-Champaign, Urbana, IL 61801, USA. [5]High Energy Physics Division, Argonne National Laboratory, Lemont, IL 60439, USA. [6]Center for Nanoscale Materials, Argonne National Laboratory, Argonne, IL 60439, USA. ✉e-mail: axelh@illinois.edu; kabjin@kaist.ac.kr; novosad@anl.gov; yili@anl.gov

control of hybrid magnonic states[28,29] beyond simple addition or subtraction of magnon excitations, and has also paved the way for quantum magnonic control such as magnon-qubit entanglement[30] and single-shot detection of individual magnons[31].

One fundamental question for cavity magnonics is whether magnons can maintain phase coherence and perform interference operations during the transduction process. Despite the demonstration of mode splitting in the frequency spectrum using continuous-wave microwave drives, studies on the real-time transduction of short magnon pulses remain scarce and are fundamentally different because the spatially nonlocal energy transduction may lead to additional decoherence. An ideal platform for investigation is a system of two remotely coupled magnon resonators[32–38]. In this configuration, both magnonic resonators are dispersively coupled to a shared microwave cavity, and magnon-magnon coupling is achieved via virtual photon exchange with the microwave cavity. This distributed hybrid magnonic system offers great potential for exploring macroscopic quantum states generation involving large ensembles of spins as well as constructing spatially distributed coherent magnonic network for interference operations[39–43].

In this work, we demonstrate time-domain multi-magnon-pulse interference in two remotely coupled yttrium iron garnet (YIG) spheres integrated on a coplanar superconducting resonator circuit[36] (Fig. 1a). Using two vertical antennas (Fig. 1b, c), we independently excite the uniform magnetostatic magnon mode in one YIG sphere and detect the response in the other, with minimal crosstalk to the superconducting resonator. With single-pulse excitation, we observe Rabi-like magnon oscillations where the magnon excitation is coherently transferred back and forth between the two YIG spheres. With two pulses, we show nearly perfect constructive or destructive interference, depending on whether the excitations are in-phase or out-of-phase, respectively, as determined by the time delay. By additionally controlling the frequency detuning between the microwave pulse and the

magnon eigenmode, we can achieve arbitrary hybrid magnonic states, ranging from constructive or destructive interference, to in-phase or out-of-phase single hybrid modes without energy transduction, enabling programmable control of hybrid magnonic states. In addition, we show diffraction-grating-like interference pattern using up to four consecutive magnon pulses, where the interference peak becomes increasingly enhanced and sharpened with the number of pulses. Our results experimentally show that magnons can preserve full coherence while coherently transferring between remotely coupled magnonic resonators, which lays a foundation for coherent information processing with hybrid magnonics and advancing functionalities in quantum magnonics.

## Strong magnon-magnon coupling

Figure 1d shows the continuous-wave measurements of microwave transmission between the two vertical antennas ($S_{21}$) under a global field of $\mu_0 H_B = 0.2$ T. A NbTi superconducting coil applies a local magnetic field to YIG sphere 1[36] with the field direction parallel to the global field. By sweeping the coil current, a clear mode anticrossing is observed between the two magnon modes, with $I_{coil} = -0.4$ A which satisfies the frequency degeneracy condition for the two magnon modes ($\omega_{m1}/2\pi = \omega_{m2}/2\pi = \omega_m/2\pi = 5.405$ GHz). A nonzero detuning field is needed to compensate the finite magnetocrystalline anisotropy in the YIG spheres with different easy axis orientations. The magnon-magnon avoided crossing yields a coupling strength of $g_{mm}/2\pi = 14.8$ MHz. This is mediated by the dispersive coupling of both YIG spheres to the nearest superconducting resonator mode ($\omega_r/2\pi = 5.27$ GHz)[39,42,44,45], with a magnon-photon coupling of of $g_{mr}/2\pi = 46$ MHz. From the frequency detuning, the theoretical magnon-magnon coupling strength is $g_{mr}^2/|\omega_m - \omega_r|2\pi = 16.3$ MHz[36], which is close to the experimental value of $g_{mm}$. We note that it is also possible to achieve remote coupling via non-resonating propagating microwave photons[46], but this will also increase magnon damping via magnon-propagating photon coupling[47]

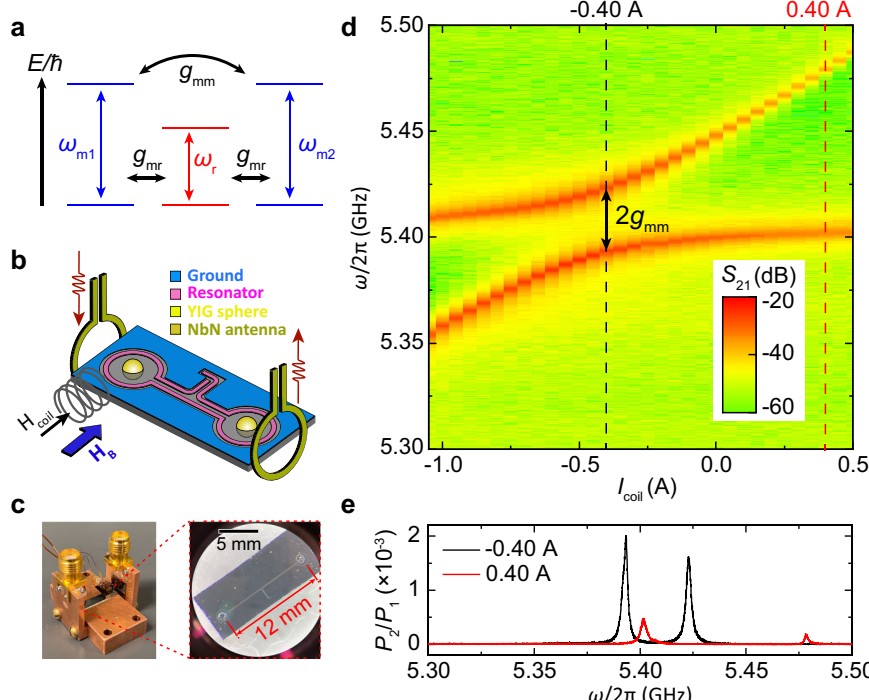

**Fig. 1 | Experiment configuration and strong magnon-magnon coupling.**
**a** Illustration of magnon-magnon coupling ($\omega_{m1}$, $\omega_{m2}$) via dispersive coupling to a microwave resonator ($\omega_r$). **b** Experimental schematics for the transmission measurement, with two vertical antennas adjacent to the two distant YIG spheres for selective microwave excitation and detection. **c** Pictures of the circuit setup and the superconducting resonator chip with two embedded YIG spheres ($d = 250$ μm) separated by 12 mm. **d** Power transmission spectra from one vertical antenna to the other vertical antenna as a function of $I_{coil}$. **e** Power spectral traces for $I_{coil} = -0.4$ A and 0.4 A. The amplitude is plotted as $P_2/P_1 = 10^{(S_{21}/10)}$.

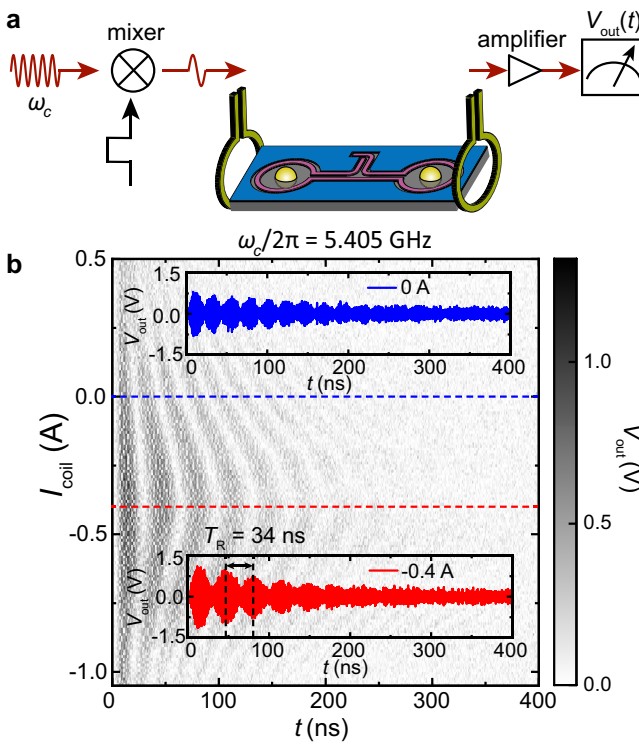

**Fig. 2 | Time-domain Rabi-like oscillation between two remotely couple YIG spheres. a** Schematic illustration of real-time measurements use microwave pulse, where the pulse input is generated by mixing a continuous microwave to a square function with a width of 10 ns, and the output is measured by a fast oscilloscope without any averaging (single-shot measurement). **b** Time traces of the output signals for different $I_{coil}$ under a global field of $\mu_0 H_B = 0.2$ T. Inset: Time trace measured at $I_{coil} = 0$ A and $I_{coil} = -0.4$ A, respectively.

and may undermine the time-domain characterization. Shown in Fig. 1e, we also highlight that the microwave transmission is significantly enhanced when the two magnon modes are degenerate and strongly coupled ($I_{coil} = -0.4$ A), as compared with the case when they are decoupled ($I_{coil} = 0.4$ A). The enhanced microwave transmission is primarily due to coherent magnon-magnon energy transduction via the superconducting resonator, whereas in the decoupled case the transmitted signal arises from free-space radiation between the two vertical antennas. The large signal-to-noise ratio of magnon excitations is crucial for observing the real-time evolution of magnon states.

## Real-time measurements of coherent magnon-magnon transduction

To investigate the temporal dynamics of the two coupled magnonic resonators, we excite the magnon mode of YIG sphere 1 with a microwave pulse from the first vertical antenna. The transmitted magnon signal to YIG sphere 2 is then coupled to the second vertical antenna, amplified by a low-noise amplifier, and then detected by a fast oscilloscope with single-shot measurements (Fig. 2a). The pulse duration is 10 ns and is much shorter than the magnon-magnon transduction period $\pi/g_{mm} = 33.8$ ns, so that the magnon in sphere 1 is excited before the transduction start to take place. In Fig. 2b, we show the measured magnon amplitude evolution of sphere 2 as a function of $I_{coil}$ with the microwave pulse frequency $\omega_c$ equal to the magnon eigenfrequency, $\omega_c/2\pi = 5.405$ GHz. A clear Rabi-like oscillation is observed, showing coherent energy transduction between the two YIG spheres for long duration. At $I_{coil} = -0.4$ A where the two magnon modes are degenerate, we find that the magnon population oscillates with a period of $T_R = 34$ ns, agreeing well with the calculated $\pi/g_{mm}$. A relaxation time $T_1 = 161.50$ ns is extracted from the exponential decay

of the oscillating amplitude. It corresponds to a magnon damping rate of $\kappa_m/2\pi = 1/(2\pi T_1) = 0.98$ MHz and agrees with the magnon linewidths measured in Fig. 1e (−0.40 A, black curve). When the magnon frequencies are detuned, e.g., at $I_{coil} = 0$ A, the magnon oscillation period becomes shorter which is due to the larger frequency difference between the two hybrid modes. We have also conducted the measurement by fixing $I_{coil} = -0.4$ A and changing $\omega$; see the Supplementary Material for details. A fixed Rabi-like oscillation period of $T_R$ is obtained, and the excitation is efficient when $\omega$ is within a bandwidth of 0.1 GHz, corresponding to the frequency broadening by the pulse width of 10 ns. The center of the band is located at $\omega = \omega_m$, showing that the pulse efficiently excites the magnon eigenmode of a single YIG sphere at $\omega_m$ instead of the eigenmode of the hybrid dynamics at $\omega_m \pm g_{mm}$.

## Two-pulse interference

To investigate magnon interference between the two remotely coupled YIG spheres, we inject two consecutive microwave pulses of equal amplitude to excite YIG sphere 1 (Fig. 3a) under the condition of $I_{coil} = -0.4$ A and $\mu_0 H_B = 0.2$ T ($\omega_{m1,2} = \omega_m$). The pulse width for the two-pulse experiments is set to 4 ns to minimize state leakage. Since both pulses are generated from the same continuous microwave source, they maintain full phase coherence, with the phase delay determined by the frequency and time delay. First, we set the microwave pulse frequency equal to the magnon frequency, as $\omega_c/2\pi = \omega_m/2\pi = 5.405$ GHz. Figure 3b shows the time evolution of two-pulse interference for different delay time $\Delta\tau$. The diagonal boundary marks the onset of the second pulse. Before the arrival of the second pulse, magnon excitation in YIG sphere 2 shows Rabi-like oscillation with a period $T_R$, consistent with the measurements in Fig. 2b. After the second pulse, a distinct interference pattern emerges, showing near-perfect construction or destruction of the Rabi-like oscillation. Specifically, the time traces show a maximal amplitude which is twice that of the single-pulse excitation at $\Delta\tau = 2nT_R$, and a near-zero amplitude at $\Delta\tau = (2n+1)T_R$ (Fig. 3c). Note that $\Delta\tau = 0$ corresponds to a single 4-ns pulse with double the amplitude. Then, we detune the microwave pulse frequency to $\omega = \omega_m + g_{mm}$. Shown in Fig. 3d, the interference pattern changes drastically. The period is halved, with constructive interference now occurring at $\Delta\tau = 2n(T_R/2)$ (Fig. 3e). Moreover, at $\Delta\tau = (2n+1)(T_R/2)$, the Rabi-like oscillations vanish, and the magnon excitation shows a monotonous decay, yielding the same $T_1$ as measured in Fig. 2b. A same feature is observed in the case of $\omega = \omega_m - g_{mm}$; see the Supplementary Information for more details. We note that the baseline of the time traces are different when $\omega$ is changed (e.g., Fig. 3b, d, and the offset between blue and green curves in Fig. 4e). We attribute it to the varying responses of the microwave circuits, including amplifier, perpendicular antenna, band-pass filter and attenuator, as a function of pulse carrier frequencies. This is a circuit artifact and does not change the results of magnon interference (see the Supplementary Information for details).

The coupled magnon dynamics can be analytically described by the time evolution of two coupled magnon resonators $\vec{m_1}$ and $\vec{m_2}$. Their hybrid eigenmodes can be formulated as:

$$\vec{m_\pm}(t) = \frac{1}{\sqrt{2}}(\vec{m_1} \pm \vec{m_2})e^{-i(\omega_m \pm g_{mm})t} \qquad (1)$$

where $\vec{m_+}$ and $\vec{m_-}$ denote the in-phase and out-of-phase hybrid modes, respectively, with eigenfrequencies $\omega_m \pm g_{mm}$. The magnon state can be labeled by the amplitudes of the two eigenmodes, $(m_+, m_-)$. After the first microwave pulse, the magnon state evolves from the initial state $(0,0)$ to the state $(1,1)$ renormalized as the maximal amplitude of each eigenmode from one single pulse. After the second pulse, the new magnon excitation will interfere with the magnon states from the first pulse as described by Eq. (1). The resulting

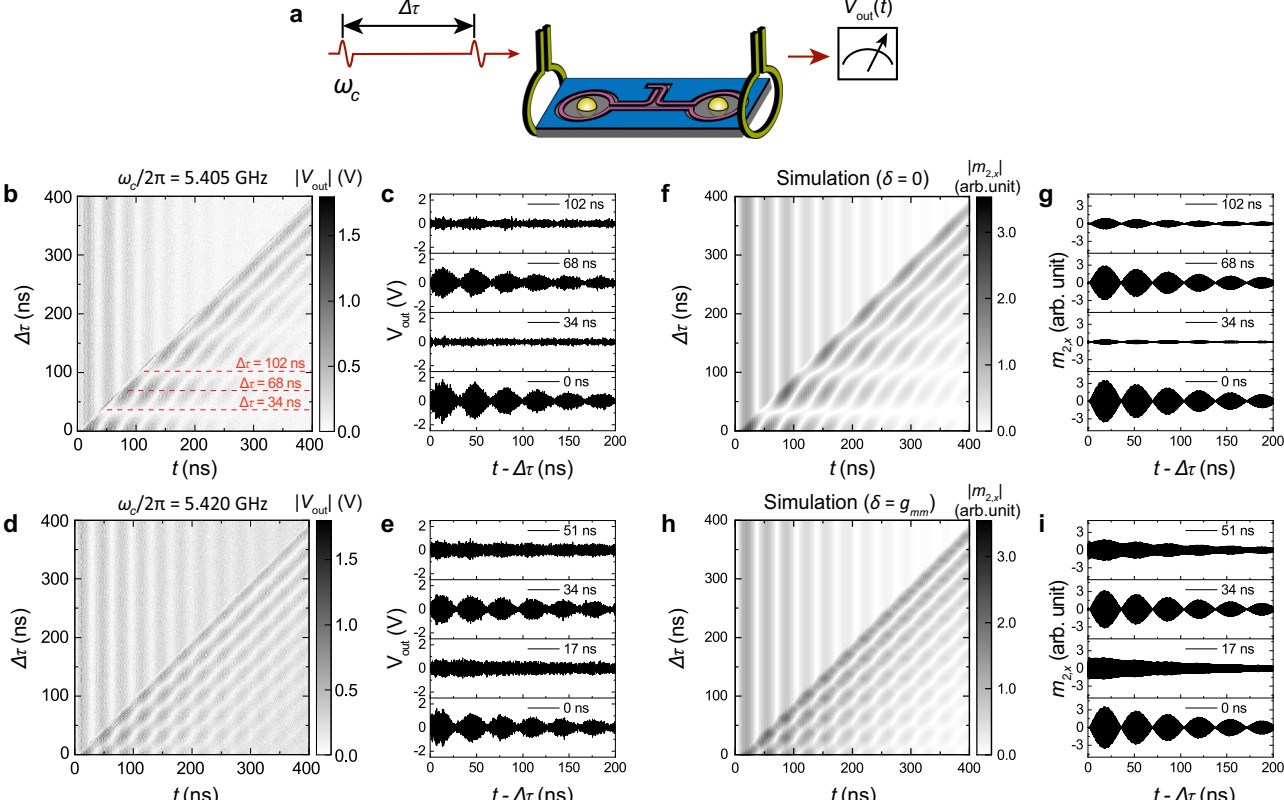

**Fig. 3 | Two-pulse magnon interference in the time domain. a** Schematic of two-pulse excitations of coherent magnonic states. **b** Time traces of the output signals for different delay time $\Delta\tau$ under two-pulse excitations, measured at $\omega_c = \omega_m$ ($\delta = 0$). **c** Individual time traces of **b** at $\Delta\tau = 0$, 34, 68 and 102 ns, with $x$-axis shifted by $\Delta\tau$.

**d** Same as **b** measured at $\omega_c = \omega_m + g_{mm}$ ($\delta = g_{mm}$). **e** Individual time traces of **d** at $\Delta\tau = 0$, 17, 34 and 51 ns. **f–i** Theoretical prediction of **b–e** using coupled harmonic oscillator model corresponding to Eq. (2) and by adding a magnon exponential decay term, $\exp(-t/T_1)$.

magnon amplitudes after the second pulse can be derived as:

$$m_\pm(\Delta\tau) = |1 + e^{-i(\delta \mp g_{mm})\Delta\tau}| = 2\left|\cos\frac{(\delta \mp g_{mm})\Delta\tau}{2}\right| \quad (2)$$

where $\delta = \omega - \omega_m$ is the frequency detuning between the microwave pulse and the magnon eigenmode. In the case of $\delta = 0$, Eq. (2) is simplified to $m_\pm(\Delta\tau) = 2|\cos(g_{mm}\Delta\tau/2)|$, which leads to constructive (2,2) or destructive (0,0) interference at $g_{mm}\Delta\tau = 2n\pi$ (0 ns, 68 ns, etc.) or $g_{mm}\Delta\tau = (2n+1)\pi$ (34 ns, 102 ns, etc.), agreeing well with the experiment in Fig. 3c. In the case of $\delta = g_{mm}$, Eq. (2) becomes $m_+ = 2, m_-(\Delta\tau) = 2|\cos(g_{mm}\Delta\tau)|$. This yields a constant amplitude of $m_+$ and an oscillating $m_-$ with half the oscillation period between (2,2) and (2,0), again consistent with Fig. 3e. Similarly, when $\delta = -g_{mm}$, according to Eq. (2), the magnon states will oscillate between (2,2) and (0,2); see the Supplementary Materials for details. In Fig. 3f–i we plot the theoretical prediction of the magnon interference pattern using two coupled harmonic oscillator model and by adding an additional exponential decay term, $\exp(-t/T_1)$ to account for magnon relaxation. These theoretical results show excellent agreement with the experimental data in Fig. 3b–e.

To further analyze the frequency components of magnon interference, we perform Fast-Fourier Transform (FFT) on the two-pulse-interference time traces. Figure 4a, b show the spectra which correspond to the time traces in Fig. 3c, e, respectively. The peaks can be clearly separated by the central magnon frequency $\omega_m/2\pi = 5.405$ GHz and correspond to the eigenfrequencies of the two hybrid mode, $\omega_m \pm g_{mm}$. Thus, the two peak amplitudes indicate the magnon final states ($m_+$, $m_-$). We note the presence of a weak, random drift in the FFT peak positions, which is likely an artifact introduced during the

FFT processing. This drift can be compensated by applying a constant spectral shift; see the Supplementary Materials for more details. The full color map of the compensated FFT spectra are shown in Fig. 4c, d as a function of $\Delta\tau$, where the peak intensity evolution agrees with the prediction in Eq. (2). In addition, we extract the peak amplitude of the $\omega_+$ and $\omega_-$ modes and plot their dependencies on $\omega_c$ and $\Delta\tau$ in Fig. 4c, d, respectively. Clear interference patterns are observed and nicely match with the calculation of Eq. (2) as shown in Fig. 4e, f. These results demonstrate that the magnon interference state ($m_+$, $m_-$) can be continuouslt and programmably controlled between (0,0) and (2,2) by adjusting the excitation frequency $\omega_c$ and the pulse delay $\Delta\tau$.

Magnon interference also allows us to quantify the magnon dephasing time $T_2$ during remote magnon transduction. Different from $T_1$ which characterizes the relaxation time of magnon states, $T_2$ describes how long magnons preserve their phase correlation for interference. The two-pulse experiments set the condition for magnon-magnon interference with a variable delay time $\Delta\tau$. The external microwave source, which is used to create the two consecutive pulses, provides a stable phase reference for analyzing magnon interference. In Fig. 4g, we plot the maximal output voltage amplitude after the second pulse, $V_{out}^{max}$, as a function of $\Delta\tau$ for $\delta = 0$ (red) and $\delta = \pm g_{mm}$ (green and black). $V_{out}^{max}$ corresponds to the sum $m_+ + m_-$ normalized to the maximal excitation amplitude. The dephasing process can be modeled by including a dephasing term in Eq. (2):

$$m_\pm(\Delta\tau) = |1 + e^{-i(\delta \mp g_{mm})\Delta\tau - \Delta\tau/T_2}| \quad (3)$$

The fitting curves to Eq. (3) nicely reproduce the experimental results for all the three frequency detunings. The extracted dephasing time, $T_2 = 139.0$ ns, is slightly shorter than the relaxation time $T_1 = 161.5$ ns,

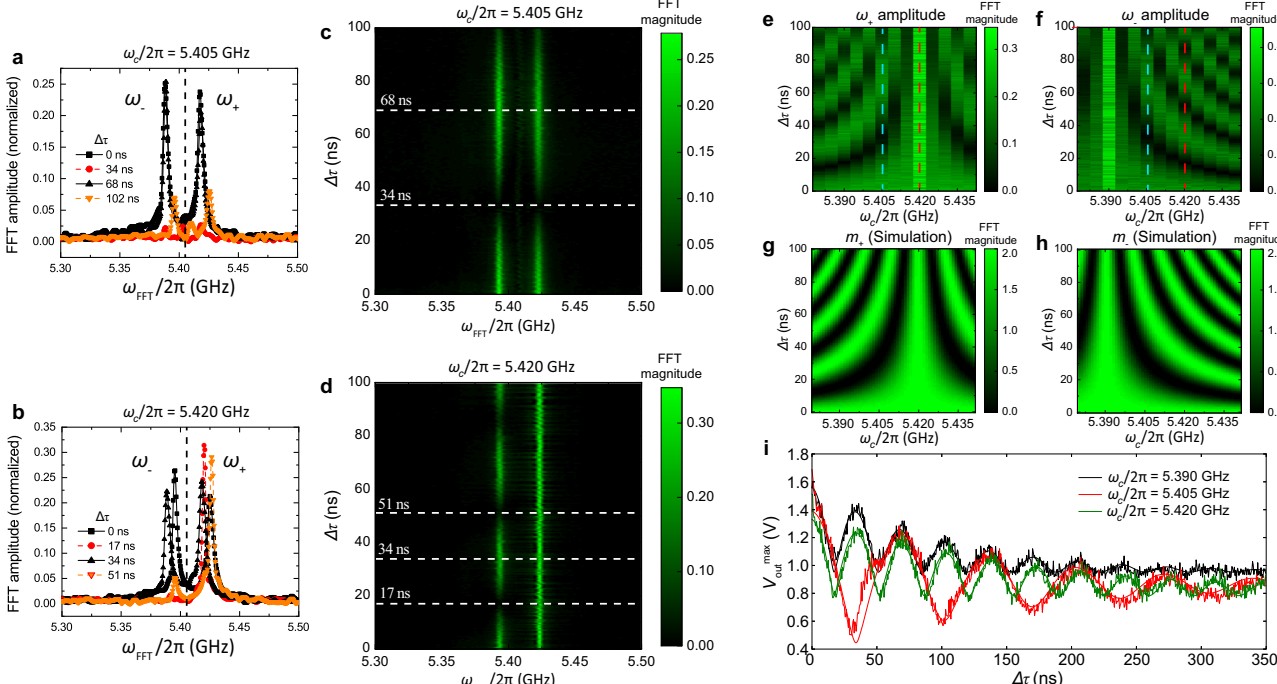

**Fig. 4 | FFT analysis and magnon dephasing time measurement. a, b** FFT spectra of two-pulse magnon interference time traces for $\delta = 0$ and $\delta = g_{mm}$ (Fig. 3c, e), respectively. The vertical dashed line denote $\omega_{FFT}/2\pi = 5.405$ GHz. **c, d** Full FFT spectra of **a, b** as a function of $\Delta\tau$. **e, f** Extracted FFT peak amplitudes of $\omega_+$ and $\omega_-$ modes as a function of $\omega_c$ and $\Delta\tau$. **g, h** Calculations of $m_+$ and $m_-$ from Eq. (2) using $g_{mm}/2\pi = 14.3$ MHz. **i** Plot of $V_{out}^{max}$ as a function of $\Delta\tau$ for $\omega/2\pi = 5.390, 5.405$ and 5.420 GHz. The fit equation is $V_{max} = V_0[m_+(\Delta\tau) + m_-(\Delta\tau)]$ by taking $V_0, g_{mm}$ and $T_2$ as fit parameters.

suggesting that the magnon spin dynamics remain highly phase coherent due to their exchange coupling. The two-pulse interference process closely resembles Ramsey interference of a single spin with two $\pi/2$ pulses[48]. We note that two-pulse magnon interference has been previously conducted in a cavity magnonic system[29,49]. However, since the results were taken by averaging many measurements, the work only showed partial destructive interference, which is likely due to phase drift during averaging. In contrast, our demonstration of single-shot, full-contrast destructive interference is essential for validating the feasibility of coherent magnon gate operations, such as Ramsey interference and active reset of magnon state, which have been widely used in quantum information science[48,50].

## Interference pattern with four magnon pulses

Finally, we demonstrate interference patterns with four magnon pulses (Fig. 5a). To show their concerted interference, we maintain a constant time delay $\Delta\tau$ between each two pulses. Figure 5b, c shows the resulting interference pattern for $\delta = 0$ and $\delta = g_{mm}$, respectively. For $\delta = 0$, constructive interference occurs at $\Delta\tau = 2nT_R$, consistent with the two-pulse case. For $\delta = g_{mm}$, the interference period is halved, as observed in the two-pulse interference in Fig. 3d. Here the measured $T_R = \pi/g_{mm} = 15.2$ ns agrees well with the magnon-magnon coupling strength $g_{mm}/2\pi = 33$ MHz. This coupling strength is measured at a different magnetic field ($\mu_0 H_B = 0.23$ T), with $I_{coil}$ set to the magnon degeneracy condition ($\omega_{m1,2}/2\pi = 6.585$ GHz). The coupling is mediated by a high-order resonator mode. However, as the number of pulses increases from two to four, the interference patterns become increasingly stronger and sharper for both detuning cases. This is a result of multi-pulse interference, which can be formulated analogous to diffraction grating interference[51]:

$$m_\pm(\Delta\tau) = \left| 1 + \sum_{n=1}^{N-1} e^{-in\Delta\phi_\pm} \right| = \frac{\sin(N\Delta\phi_\pm/2)}{\sin(\Delta\phi_\pm/2)} \qquad (4)$$

where $N$ is the number of pulses and the phase delay is defined as $\Delta\phi_\pm = (\delta \mp g_{mm})\Delta\tau$. Constructive interference happens when the phase delay satisfies the grating equation: $\Delta\phi_\pm = 2\pi k$ where $k = 0, 1, 2...$ As $N$ is increased, the interference becomes both stronger and narrower. Figure 5d, e shows the theoretical predictions from Eq. (4) including an exponential magnon decay term, agreeing well with the experiments in Fig. 5b, c. In addition, we shows the maximal output amplitudes in Fig. 5f, g with 2, 3 and 4 pulses. For $\delta = 0$, the experimental result agrees well with the theoretical prediction from Eq. (4) as plotted in Fig. 5h. For $\delta = g_{mm}$, comparing with the theoretical prediction in Fig. 5i, the minimal output amplitude during destructive interference is floored at the single-pulse level (blue), consistent with the theoretical expectation. Nevertheless, the maximal output amplitude decays differently from the prediction using $T_2 = 139$ ns. This deviation suggests a more complicated dephasing process during multi-pulse interference including phase slips and nonlinear behaviors of the mixer, which require further investigations. These results show that the magnon interference using multiple microwave pulses can be scaled to large pulse numbers, indication the potential of our hybrid magnonic device for implementing highly complex sequences of time-domain gate operations.

In conclusion, we have demonstrated single-shot real-time interference between two remotely coupled magnon resonators (YIG spheres) that are embedded in a coplanar superconducting resonator. By precisely controlling the frequency detuning and time delay of microwave pulses, we demonstrate coherent magnon interference with single-shot measurements, including both constructive and destructive magnon interference for two pulses, and multi-pulse interference up to four pulses showing diffraction-grating-like behaviors. From the time-delay dependence of magnon interference patterns, we extract the magnon dephasing time, which closely matches the magnon relaxation time. Our results establish a realistic, circuit-integrated hybrid magnonic platform for implementing coherent magnon operations in the time domain. The ability to control

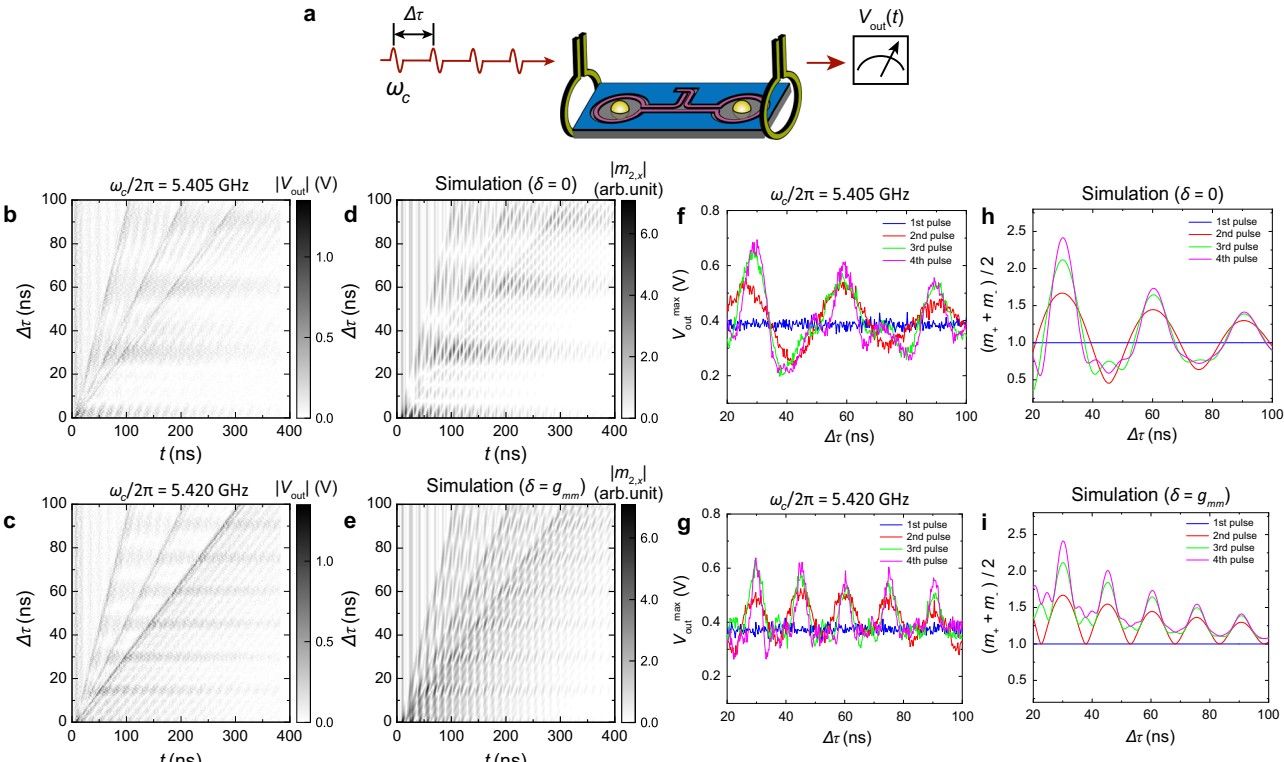

**Fig. 5 | Magnon interference with four pulses. a** Schematic of four-pulse excitations of coherent magnonic states. Time traces of the output signals for different $\Delta\tau$ under two-, three- and four-pulse excitations, measured at **b** $\delta = 0$ and **c** $\delta = g_{mm}$. **d**, **e** Theoretical prediction of **b**, **c** using coupled harmonic oscillator model corresponding to Eq. (4) by adding an exponential decay term $e^{-t/T_1}$ with $T_1 = 161.5$ ns. **f**, **g** Maximal output voltage amplitudes after 2, 3 and 4 pulses extracted from **b**, **c**, respectively. **h**, **i** Theoretical prediction of maximal output voltage from Eq. (4) by adding an exponential decay term $e^{-t/T_2}$ with $T_2 = 139$ ns. Here, $\omega_m/2\pi = 6.585$ GHz and $g_{mm}/2\pi = 33$ MHz are measured at $\mu_0 H_B = 0.23$ T.

hybrid magnon states opens up pathways for realizing remote magnon-magnon entanglement and distributed magnon-based quantum gate operations through integrating superconducting qubits on the same superconducting circuit[31,52,53]. Especially, our results show the capability of performing multiple time-domain gate operations within a hybrid magnonic system. Additionally, the two-magnon-resonator system can be extended towards coupled magnon networks with distributed magnon oscillators, which is applicable to many magnonic computing concepts with ultra-long magnon coherence time, tunable magnon frequency and coupling strength.

## Methods
### Film preparation and device fabrication
The superconducting coplanar waveguide (CPW) resonator was patterned by photolithography and reactive ion etching from 200-nm NbN film sputtered on a high-resistance Si substrate. YIG spheres were mounted in the holes at the center of the loop antenna, and the holes were etched on Si substrate with a depth of 125 μm and a diameter of 250 μm which exactly fit to the dimension of the YIG spheres. Deep holes were etched by reactive ion etching (RIE). To mount the YIG sphere, a small amount of diluted GE varnish was put into each hole, followed by the prompt insertion of the YIG sphere. The substrate with NbN CPW resonator and two YIG spheres was put on a copper block and fixed using the diluted GE varnish. A local NbTi superconducting coil was made by hand-winding on a polyethylene rod and integrated adjacent to one YIG sphere. Two vertical antennae were patterned as loop-shaped with a diameter of 2 mm using the same fabrication process as for the CPW resonator, and assembled at the side of the two YIG spheres with SMA connectors.

### Measurement details
The measurements were performed at a temperature of 1.5 K, using a 3-axis superconducting magnet cryostat system manufactured by American Magnetics Inc. The global dc magnetic field is set to be 0.2 T for all the experiments shown in this paper. A Keysight PNA vector network analyzer (VNA) was used to measure the system's microwave transmission spectroscopy. An input power and an IF bandwidth for the VNA measurement were set to be −50 dBm and 200, respectively. A Lecroy SDA 18000 real-time oscilloscope was used for the time-domain measurements, using an 11 GHz bandwidth mode with a sampling rate of 20 GSa/s. To amplify the voltage and enhance the signal-to-noise ratio of the microwave signal, two 30 dB-low-noise RF amplifiers (ZVA-183Gx-S+) and a highpass filter (VHF-2275+, 2640-6230 MHz) manufactured by MiniCircuit were used before the signal from the device is injected into the oscilloscope. For the single-pulse measurements, the microwave pulses were generated by mixing 10 ns-square pulse train for single pulse excitations and 4 ns-square pulse train for multiple pulse excitations (2.0 V amplitude, 1 kHz repetition rate) from a Picosecond pulse generator and a continuous microwave (0 dBm amplitude) from Agilent E8257D Analog Signal Generator, using a Pasternack RF mixer (PE86X1027). During the time domain-measurement, a trigger out signal from the Picosecond pulse generator is injected to the Lecroy oscilloscope through a few meters of microwave cable, to capture the timing of exciting the coupled oscillator system. Time delay of multiple pulses were generated by Stanford Research DG645 Digital Delay/Pulse Generator. The time delay $\Delta\tau$ in this paper refers to the time between the left edges of multiple square pulses, so $\Delta\tau = 0$ means 4-ns single pulse with double amplitude.

## Data availability

The datasets generated and/or analyzed during the current study are available from the corresponding authors on reasonable request.

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

## Acknowledgements

We thank Vasyl Tyberkevych and Cody Trevillian for suggesting the idea of multi-pulse interference, as well as Wolfgang Pfaff, Juliang Li, Volodymyr G. Yefremenko and Margarita Lisovenko for discussion and support on the experiment. This work was supported by the U.S. DOE, Office of Science, Basic Energy Sciences, Materials Sciences and Engineering Division under contract No. DE-SC0022060. The lithographic patterning and fabrication of the superconducting resonators performed at the Center for Nanoscale Materials, a U.S. Department of Energy Office of Science User Facility, was supported by the U.S. DOE, Office of Basic Energy Sciences, under Contract No. DE-AC02-06CH11357. K.-J.K. is supported by KAIST-funded Global Singularity Research Program for 2021 and the National Research Foundation of Korea (NRF) funded by the Korean Government (MSIP) under grant RS-2023-00275259, RS-2023-00207732. M.S. was supported by the education and training program of the Quantum Information Research Support Center, funded through the National research foundation of Korea (NRF) by the Ministry of Science and ICT (MSIT) of the Korean government under grant No. 2021M3H3A103657313.

## Author contributions

M.S., V.N. and Y.L. conceived the idea, conducted the experiment, and developed the theoretical analysis, T.P. grew the NbN film, T.W.C. assisted the Si etching for YIG sphere mounting, R.D. assisted the resonator fabrication, U.W. and W.K.K. assisted the cryogenic measurements, J.L. assisted the data analysis, J.P. assisted the experiment configuration, A.H., K.J.K., V.N. and Y.L. supervised the entire experiment. All authors contributed to the result discussion and manuscript writing.

## Competing interests

The authors declare no competing interests.
