## [Transparent Peer Review file · Nature Communications]

Single-shot magnon interference in a magnon-superconducting-resonator hybrid circuit

Corresponding Author: Dr Yi Li

Version 0:

Reviewer comments:

Reviewer #1

(Remarks to the Author)

Please find attached our report.

Reviewer #2

(Remarks to the Author)

This work demonstrates the precise control of time interference between two magnon states using a pulse sequence, which is important for advancing coherent information processing in hybrid magnonic systems. However, due to concerns about the accuracy of certain statements in the manuscript, I cannot recommend the publication of the paper in its current form.

First of all, in order not to mislead readers, I suggest that the authors avoid using the word 'diffraction', as no actual diffraction occurs in their experiment. Additionally, the authors stated that they conducted an FFT of the time traces in Fig. 3(c) and (e) and plotted the results in Fig. 3(j) and (k). However, these frequency spectra differ significantly from those shown in Fig. 4(a) and (b). For example, at $\Delta\tau=51$ ns, multiple peaks are visible in Fig. 4(a), whereas only two peaks appear in Fig. 3(k). Did the authors measure the two pulse signals at zero magnetic field and then convert these signals into the frequency domain using the FFT method? I suspect they might obtain results similar to those in Fig. 4(a) and (b) under these conditions. This would imply that the 'diffraction grating' arises solely from time interference between two consecutive pulses, rather than from interference between two magnon states as the authors propose.

Reviewer #3

(Remarks to the Author)

After reviewing the revised manuscript, I am pleased to confirm that the authors have addressed all the concerns and suggestions raised in my previous review. I now consider the manuscript to be of publication quality and recommend it for acceptance.

Report of Reviewer 1:

[Previously, our main critics were at the interpretation of the data as a truly interference phenomenon, and the overall novelty of the experiment. The authors have replied our critics elaborating more about our raised concerns on cross-talk effects, which are indeed not negligible. The response was very detailed regarding our conceptual questions; the FFT traces shown in Fig. 3 (j,k) support the conclusion that the time dynamics is indeed an interference effect. Furthermore, the new version adds several information about the system (e.g. detuning is required to compensate magnetocrystalline anisotropy) which further elucidate the physics and results presented. We would also like to acknowledge the added theory, which further supports the conclusion presented in the paper, and the other text modifications that follow our suggestions and the suggestions of the other referees.]

We sincerely thank the reviewer for acknowledging the novelty of the results as well as all the constructive suggestions in the previous round of review.

[We agree that the demonstration of a single-shot time domain interference for this system is interesting, but we disagree with the initial statement in the rebuttal letter that anti-crossing is only a good indicator of strong coupling. Anti-crossing is an unequivocal indicator of strong coupling. Avoided crossings are obtained via time-domain signals that exhibit interference effects. It is true that, in most studies, one average over many time-domain traces to get statistics. In this sense, any demonstration of avoided crossing is intimately related with the presence of interference effects. We would, nevertheless, like to emphasize that our disagreement with the authors statement does not diminish the relevance of the results shown in the manuscript.]

The reviewer is right about the point that anti-crossing is an unequivocal indicator of strong coupling. The frequency-domain measurements is an accurate reflection of spectrum for the time-domain measurements. We apologize for the inaccuracy in our previous rebuttal letter.

The main advantage of time-domain measurements over frequency domain lies in the accurate measure of delay-related phase evolution which is the key for real-time interference. In a VNA, the phase of microwave transmission can be also measured but only in the continuous-wave state. Certainly VNA-measured avoided crossing is intimately related with the interference effect, but it is only equivalent to interference with continuous excitation. In our work we focus on magnon interference with short-duration pulse excitation, which is more relevant to coherent information processing and ultimately to a future prototype of quantum magnonics.

[Overall, our evaluation is that the manuscript is now more clear and well presented, and the conclusions are now better supported and discussed in the text. We have some questions that might be addressed before the publication of the manuscript:

1. In Fig. 3j there is always a frequency component at the middle of the hybrid modes which does not

seems to be a measurement artifact (since it is not present in the other FFT plot). This is also not the resonator frequency. Since the experiment exhibits a cross-talk between the antennas, could this frequency component be a consequence of the cross-talk?

2. How well the FFT of the theory time-signal matches the measured data? It could be helpful to have those (maybe in the appendix).]

We thank the reviewer for the constructive question. To address the question, we have added the FFT of Fig. 3b and d, as the new Fig. 4c and d, in our resubmitted main text. This has also been requested by Reviewer 2. The new FFT clearly shows that the magnon interference is originated from the two hybrid magnon modes (ω_+ and ω_-), and the amplitude of the ω_+ and ω_- modes change with $\Delta\tau$ following the prediction of Eq. (2) of the main text. This should address question 2 above.

For question 1, in our new FFT spectra, there is indeed a weak peak located at 5.405 GHz, which can be attributed to a finite cross talk between the two vertical antennas. We have added the discussion in the Supplemental Materials.

Figure R-1: FFT color map of 2-pulse interference as a function of $\Delta\tau$ for $\omega_c/2\pi = 5.405 \text{ GHz}$. This figure is shown as Fig. 4(c) of the main text and Fig. S4(b) of the Supplemental Materials.

In addition, in the Supplemental Materials, we have added the discussion about a random frequency drift at different $\Delta\tau$ during the measurements. This random frequency drift is an artifact of the pulse microwave measurements and FFT processing, and is unrelated to any physics in the hybrid magnon dynamics.

Report of Reviewer 2:

[This work demonstrates the precise control of time interference between two magnon states using a pulse

sequence, which is important for advancing coherent information processing in hybrid magnonic systems. However, due to concerns about the accuracy of certain statements in the manuscript, I cannot recommend the publication of the paper in its current form.]

We thank the reviewer for the positive comment about our work. Below we have addressed the concerns about the statement and terms used in the manuscript

[First of all, in order not to mislead readers, I suggest that the authors avoid using the word 'diffraction', as no actual diffraction occurs in their experiment.]

We agree with the referee that the word 'diffraction' is confusing because it usually describes interference from a continuous phase evolution, usually ending up as a sinc function. We have carefully examined the manuscript, and have removed a few individual use of 'diffraction'. However, we point out that the term 'diffraction grating' is a well-accepted physical term, with an analytical amplitude expression identical to multi-magnon interference [Eq. (4) in the main text, note that the intensity expression should take a square, but the oscilloscope only measures the amplitude]:

$$m_{\pm}(\Delta\tau) = \left| 1 + \sum_{n=1}^{N-1} e^{-in\Delta\phi_{\pm}} \right| = \frac{\sin(N\Delta\phi_{\pm}/2)}{\sin(\Delta\phi_{\pm}/2)} \quad (1)$$

This equation can be used to nicely reproduce the multi-pulse interference in Fig. 5. Thus, we think it is reasonable to highlight the similarity of our work to 'diffraction grating' in the manuscript, which has a different meaning from 'diffraction' alone.

[Additionally, the authors stated that they conducted an FFT of the time traces in Fig. 3(c) and (e) and plotted the results in Fig. 3(j) and (k). However, these frequency spectra differ significantly from those shown in Fig. 4(a) and (b). For example, at $\tau = 51$ ns, multiple peaks are visible in Fig. 4(a), whereas only two peaks appear in Fig. 3(k). Did the authors measure the two pulse signals at zero magnetic field and then convert these signals into the frequency domain using the FFT method? I suspect they might obtain results similar to those in Fig. 4(a) and (b) under these conditions. This would imply that the 'diffraction grating' arises solely from time interference between two consecutive pulses, rather than from interference between two magnon states as the authors propose.]

We thank the reviewer for pointing out the issue in Fig. 4(a) and (b). Indeed it is somewhat misleading. The x-label for Figs. 4(a) and (b) should be ω_c , instead of ω_{FFT} . In another word, the colormap shows the amplitude of the ω_+ and ω_- peaks, instead of full FFT spectra as a function of $\Delta\tau$.

To address the question of the reviewer, we have added two new FFT color map plots a function of $\Delta\tau$ for $\omega_c/2\pi = 5.405$ GHz and 5.420 GHz, as shown in Fig. 4(c) and (d). It is clear that in the new FFT spectra, only two peaks are extracted at 5.390 GHz and 5.420 GHz, which correspond

to ω_- and ω_+ , respectively. Their FFT amplitude evolution with $\Delta\tau$ also agrees well with the prediction from Eq. (2). This is an unambiguous evidence that the results arise from interference between two hybrid magnon states (ω_+ and ω_-).

Response to reviewer 3:

After reviewing the revised manuscript, I am pleased to confirm that the authors have addressed all the concerns and suggestions raised in my previous review. I now consider the manuscript to be of publication quality and recommend it for acceptance

We sincerely thank the reviewer for his/her positive recommendation of our revised manuscript for publication!

Previously, our main critics were at the interpretation of the data as a truly interference phenomenon, and the overall novelty of the experiment. The authors have replied our critics elaborating more about our raised concerns on cross-talk effects, which are indeed not negligible. The response was very detailed regarding our conceptual questions; the FFT traces shown in Fig. 3 (j,k) support the conclusion that the time dynamics is indeed an interference effect. Furthermore, the new version adds several information about the system (e.g. detuning is required to compensate magnetocrystalline anisotropy) which further elucidate the physics and results presented. We would also like to acknowledge the added theory, which further supports the conclusion presented in the paper, and the other text modifications that follow our suggestions and the suggestions of the other referees.

We agree that the demonstration of a single-shot time domain interference for this system is interesting, but we disagree with the initial statement in the rebuttal letter that anti-crossing is only a good indicator of strong coupling. Anti-crossing is an *unequivocal* indicator of strong coupling. Avoided crossings are obtained via time-domain signals that exhibit interference effects. It is true that, in most studies, one average over many time-domain traces to get statistics. In this sense, any demonstration of avoided crossing is intimately related with the presence of interference effects. We would, nevertheless, like to emphasize that our disagreement with the author's statement does not diminish the relevance of the results shown in the manuscript.

Overall, our evaluation is that the manuscript is now more clear and well presented, and the conclusions are now better supported and discussed in the text. We have some questions that might be addressed before the publication of the manuscript

1. In Fig. 3j there is always a frequency component at the middle of the hybrid modes which does not seems to be a measurement artifact (since it is not present in the other FFT plot). This is also not the resonator frequency. Since the experiment exhibits a cross-talk between the antennas, could this frequency component be a consequence of the cross-talk?
2. How well the FFT of the theory time-signal matches the measured data? It could be helpful to have those (maybe in the appendix).